# Crocus Sativus for Insomnia: A Systematic Review and Meta-Analysis

**DOI:** 10.3390/ijerph191811658

**Published:** 2022-09-16

**Authors:** Mohd Puad Munirah, Mohd Noor Norhayati, Mohamad Noraini

**Affiliations:** 1Department of Family Medicine, School of Medical Sciences, Universiti Sains Malaysia, Kubang Kerian, Kota Bharu 16150, Malaysia; 2School of Dental Sciences, Health Campus, Universiti Sains Malaysia, Kubang Kerian, Kota Bharu 16150, Malaysia

**Keywords:** crocus sativus, insomnia, sleep quality

## Abstract

Insomnia is a common complaint affecting human performance in daily life activities. This study aimed to analyze the effects of Crocus sativus on insomnia. Methods: PRISMA guidelines were used according to the PICOS model. A systematic search of PubMed/Medline and the Cochrane Library was undertaken for literature until December 2021. A random effects model was used with I^2^ statistic to assess heterogeneity and a GRADE assessment was used to assess the quality of the outcomes. Eight articles were included, involving 431 participants. Crocus sativus reduced insomnia severity (SMD: 0.53; 95%CI: −0.05 to 1.11; I^2^ statistic = 59%; *p* = 0.08) and increased sleep quality (SMD 0.89, 95% CI 0.10 to 1.68; I^2^ statistic = 90%; *p* = 0.03; 6 studies, 308 participants, very low-quality evidence) and duration (SMD: 0.57; 95%CI: 0.21 to 0.93; I^2^ statistic = 40%; *p* = 0.002; 5 studies; 220 participants, moderate-quality evidence) compared with the placebos. Although there is limited evidence of a very low- to moderate-quality, Crocus sativus may benefit people with insomnia. This non-pharmacological intervention may reduce the chance of adults with insomnia taking sedative–hypnotic medication, thus reducing dependency and withdrawal symptoms.

## 1. Introduction

Insomnia is one of the most common medical complaints. Insomnia may be experienced at all stages of adulthood, and the problem is chronic for millions. Furthermore, it can be a symptom of other disorders, such as depression or a primary disorder. Most often, insomnia is defined by the presence of an individual report of difficulty with sleep [1]. Insomnia that lasts for one year or more is a risk factor for the development of depression [2]. The presence of long sleep latency, frequent awakenings during nighttime, prolonged periods of wakefulness during the sleep period, or even frequent transient arousals is defined as evidence of insomnia [1].

The recent definition by the Diagnostic and Statistical Manual of Mental Disorders, Fifth Edition (DSM-5) states that it makes no distinction between primary and comorbid insomnia. The DSM-5 defines insomnia as dissatisfaction with sleep quantity or quality, associated with one (or more) symptoms such as difficulty initiating sleep, difficulty maintaining sleep, which is characterized by frequent awakenings or problems returning to sleep after awakenings, and early-morning awakening with an inability to return to sleep. According to the other criteria, sleep disturbance also causes clinically significant distress or impairments in social, occupational, educational, academic, behavioral, or other important areas of functioning. The sleep difficulty occurs at least three nights per week, is present for at least three months, occurs despite adequate opportunity for sleep, cannot be explained by, and does not occur exclusively during, another sleep–wake disorder, is not attributable to the physiological effects of a drug of abuse or medication, and coexisting mental disorders and medical conditions do not adequately explain the predominant complaint of insomnia [3].

The prevalence of insomnia in adults is approximately 10% to 20%, with approximately 50% among these having a chronic course [4]. Studies conducted in Western settings have reported that primary care populations have a higher prevalence of insomnia (64–69%) than the general population [5]. Chronic insomnia was discovered in 33% of the adult population sampled. Risk factors that are commonly associated with insomnia are increasing age and diabetes. Patients with diabetes had statistically significant insomnia compared with those without diabetes [6]. 

Herbal medicines are one of the most frequently used complementary and alternative insomnia treatments. Saffron, a spice derived from the stigmas of the Crocus sativus flower, has been confirmed in several systematic reviews and meta-analyses to be an effective natural agent for treating mild-to-moderate depression [7,8]. 

Saffron has many therapeutic effects, including diuretic, analgesic, anti-inflammatory, hepatoprotective, appetite suppressant, hypnotic, antidepressant, and bronchodilator effects [9]. Apart from being used as an herbal sedative, it has also been used as an antispasmodic, aphrodisiac, diaphoretic, expectorant, stimulant, stomachic, anticatarrhal, eupeptic, gingival sedative, and emmenagogue [10]. 

In addition, saffron and its crocin and safranal have induced hypnotic effects by increasing NREM sleep duration and decreasing its latency in animal models [11]. Saffron extract and its active compounds are known to modulate serotonin, dopamine, norepinephrine, glutamate, and GABA-A neurotransmitters. Thus, it activates the sleep-promoting neurons from the ventrolateral preoptic nucleus and inhibits the wakefulness-promoting neurons from the tuberomammillary nuclei in vitro [12]. Saffron stigma is commonly used for insomnia and anxiety in traditional medicine [9]. It has increased the duration of non-rapid eye movement sleep, shortened NREM sleep latency, and enhanced the delta power activity of NREM sleep in mice. The hypnotic effects of saffron may be related to the activation of the sleep-promoting neurons in the ventrolateral preoptic nucleus and the simultaneous inhibition of the wakefulness-promoting neurons in the tuberomammillary [12].

Poor sleep quality has significant health implications as it can have a negative impact on both mental and physical health, and can interfere with daily functioning [13]. Though saffron is widely used as herbal medicine, no well-documented study has categorized the toxic effects of saffron in animal models and human studies [14]. While common side effects of saffron consumption include nausea, dry mouth, poor appetite, and headache, no serious adverse reactions have been reported [15]. This review aims to report the effects of Crocus sativus on insomnia in adults. 

## 2. Materials and Methods

Our protocol was registered in PROSPERO with the registration number CRD42022308316. The research was conducted according to the PRISMA guidelines. Relevant studies were identified by searching the following databases: Cochrane Central Register of Controlled Trials (CENTRAL) and MEDLINE (PubMed) for randomized control trials (RCTs) not later than 31 December 2021, which compare Crocus sativus for insomnia with placebos. The studies were reviewed using the Preferred Reporting Items for Systematic Reviews and Meta-Analyses (PRISMA) 2020 guidelines [16].

### 2.1. Literature Searching Strategies

We searched the Cochrane Central Register of Controlled Trials, CENTRAL (December 2021) and MEDLINE (2010 to December 2021). We used the search strategy in Appendix A to search MEDLINE and CENTRAL. We restricted the publications to the English language only. We checked the reference list of identified RCTs and review articles to find unpublished trials or trials not identified by electronic searches. We searched for ongoing trials through the World Health Organization (WHO) International Clinical Trials Registry Platform (ICTRP), http://www.who.int/ictrp/en/ (accessed on 5 January 2022) and www.clinicaltrials.gov (accessed on 5 January 2022).

### 2.2. Inclusion and Exclusion Criteria

We included randomized control trials (RCTs) comparing Crocus sativus for insomnia with placebos in adults with insomnia. We included blinded and open-label studies. The interventions included Crocus sativus extract, either herbal oil or oral capsule. The comparisons included placebo, no treatment, or standard treatment. Insomnia must have been diagnosed by clinicians. The outcomes were divided into primary and secondary. The primary outcomes included insomnia severity and sleep quality, including subjective sleep quality, sleep latency, sleep duration, sleep efficiency, sleep disturbances, use of sleeping medications, and daytime dysfunction. The secondary outcomes included anxiety level, depression level, restorative sleep, and quality of life, such as physical and mental wellbeing. 

### 2.3. Quality Assessment

Two review authors (M.P.M., M.N.N.) scanned the titles and abstracts from the searches and obtained full-text articles when they appeared to meet the eligibility criteria or when there was insufficient information to assess the eligibility. We independently evaluated the eligibility of the trials and documented the reasons for exclusion. We resolved any disagreements between the review authors by discussion. We contacted the authors if clarification was needed. 

From each of the selected studies, we extracted: the study setting, participant characteristics (age, sex, ethnicity), methodology (number of participants randomized and analyzed, duration of follow-up), method for assessing insomnia, and occurrence of related adverse events. We assessed the risk of bias based on random sequence generation, allocation concealment, blinding of participants and personnel, blinding of outcome assessors, completeness of outcome data, selectivity of outcome reporting, and other biases [17]. We resolved any disagreements by discussion. We assessed the quality of evidence for primary and secondary outcomes according to the GRADE methodology [18] for risk of bias, inconsistency, indirectness, imprecision, and publication bias, classified as very low, low, moderate, or high.

### 2.4. Statistical Analyses

Meta-analyses were performed using Review Manager 5.4 software (RevMan 2020). We used a random-effects model to pool data. Thresholds for the interpretation of the I^2^ statistic can be misleading since the importance of inconsistency depends on several factors. We used the guide to interpret the heterogeneity as outlined: 0% to 40% might not be important; 30% to 60% may represent moderate heterogeneity; 50% to 90% may represent substantial heterogeneity; and 75% to 100% would be considerable heterogeneity [17]. 

We assessed the presence of heterogeneity in two steps. First, we assessed obvious heterogeneity at face value by comparing populations, settings, interventions, and outcomes. Second, we assessed statistical heterogeneity using the I^2^ statistic [17]. 

We measured the treatment effect for dichotomous outcomes using risk ratios (RRs) and absolute risk reduction, and for continuous outcomes, we used mean differences (MDs), both with 95% confidence intervals (CIs). The planned subgroup analyses were routes of saffron administration and the presence of comorbidity.

We checked the included trials for unit of analysis errors. Unit of analysis errors can occur when trials randomize participants into intervention or control groups in clusters, but analyze the results using the total number of individual participants. We contacted the original trial authors to request missing or inadequately reported data. We performed analyses on the available data if missing data were not available. We performed a sensitivity analysis to investigate the impact of risk of bias for sequence generation and allocation concealment in the included studies. If there were sufficient studies, we intended to use funnel plots to assess the possibility of reporting biases, small study biases, or both.

## 3. Results

### 3.1. Results of the Search

A total of 346 records were found from the comprehensive database search (Figure 1). Upon elimination of duplicates, a total of 46 records were screened for eligibility. We reviewed full copies of 22 studies; we identified 8 articles as possibly meeting the review inclusion criteria, and 5 that were not eligible for inclusion. Three were reviews [19,20,21], one trial did not include an outcome of interest [22], and in one trial the intervention was a part of the control [23]. Therefore, we included eight studies.

We included 8 studies with 431 participants [24,25,26,27,28,29,30,31] (Table 1). One study was reported in two reports (Lopresti et al., 2020; Lopresti et al., 2021) and was referred to as Lopresti et al., (2021). Five out of 8 studies (and all that contributed to the primary outcome) declared funding from pharmaceutical and nutraceutical companies [25,26,27,30,31].

Five of the eight studies were conducted in high-income countries [24,25,26,27,31], and three studies were conducted in middle-income countries [28,29,30]. Four studies recruited participants from healthcare settings [24,28,29,30], and three recruited participants from social media, posters, and local newspapers [25,27,31]. One study did not mention the setting from which the participants were recruited [26]. Two studies reported insomnia participants with an underlying comorbidity of diabetes mellitus [28,30] and six reported insomnia participants with no comorbidity [24,25,26,27,29,31]. 

Participants in the studies were randomized into intervention and control groups. For eight studies, the intervention was a single dose of saffron per day [24,26,27,28,29,30,31], and one study included two groups each receiving different saffron dosages, 14 mg and 28 mg [25]. The total doses of saffron extract used in the studies were 0.6 mg per day [26], 7.5 mg per day [24,31], 8 mg intranasally [29], 15.5 mg per day [27], 28 mg per day [25], 100 mg per day [30], and 300 mg per day [28]. In seven studies, the saffron was given via a pill or capsule [24,25,26,27,28,30,31], and in one study saffron extract in liquid preparation was administered intranasally [29]. The liquid contained three ingredients: Viola odorata L., Crocus sativus L., and Lactuca sativa L [29]. The studies were conducted for one week [28], two weeks [24,31], four weeks [25,26], six weeks [27], and eight weeks [29,30]. 

Seven studies included in this meta-analysis used interventions and placebos in capsule form [24,25,26,27,28,30,31]. In one study, the intervention and the placebo were in liquid form [29]. The liquid form placebo contained sesame oil [29]. Four studies used dextrin [24,27,30,31], and two used cellulose and calcium hydrogen phosphate [25,26]. One study did not describe the ingredient in the placebo [28]. 

Insomnia severity and sleep quality were the primary outcomes of our review. The Insomnia Severity Index and Insomnia Symptom Questionnaire assessed the severity of the insomnia. The Insomnia Severity Index is a seven-item self-report questionnaire assessing the nature, severity, and impact of insomnia [32]. It assesses the perceived severity of difficulties initiating sleep, staying asleep, early morning awakening, satisfaction with current sleep patterns, and interference with daily functioning. A 5-point Likert scale is used to rate each item yielding a total score ranging from 0 to 28, with higher scores indicating more severe insomnia [33]. 

The Insomnia Symptom Questionnaire is a 13-item self-report instrument designed to identify insomnia. The items assessed the presence of a complaint of difficulty initiating or maintaining sleep, or a feeling that the sleep was nonrestorative or unrefreshing; the frequency of complaints and the duration of these symptoms; and the severity of daytime correlates of the sleep complaint. Higher scores indicate more severe insomnia [34]. 

Sleep quality in this review was assessed using the Pittsburgh Sleep Quality Index and the Pittsburgh Sleep Diary. The Pittsburgh Sleep Quality Index is a common tool that consists of a 19-item questionnaire used to measure sleep quality complaints [35,36,37]. It is divided into seven components that assess habitual sleep duration, nocturnal sleep disturbances, sleep latency, sleep quality, daytime dysfunction, sleep medication usage, and sleep efficiency. Every seven components are scored between 0 and 3. The total scores range from 0 to 21, with higher scores indicating poorer sleep quality [36,37].

The Pittsburgh Sleep Diary is a 14-item sleep diary composed of total sleep time, sleep latency, the number of awakenings after sleep onset, sleep quality, mood, and alertness. It consists of a 5-point Likert rating ranging from very bad to very good, with higher scores indicating better sleep quality [38].

Secondary outcomes included in our review were anxiety level, depression level, restorative sleep, and quality of life. Anxiety level was reported in two trials [26,28]. Anxiety level was measured using the Profile of Mood States in one trial [26] and one trial used the Spielberg Anxiety Inventory [28].

The Profile of Mood States is a widely applied measure for assessing an individual’s mood. It consists of 35 items with 4 scales: dejection/anxiety (14 items), fatigue (7 items), vigor (7 items), and anger (7 items). Determining anxiety level using the POMS is a way to assess a patient’s current mood state, which is rated on a 4-point scale (from not at all to extremely). The higher the score, the higher the level of anxiety [39,40]. The DASS-21 is a validated self-reported measure assessing stress, anxiety, and depression symptoms. It is composed of 21 questions which are rated on a 4-point scale. Subscale scores for depression, anxiety, and stress are calculated. The higher the scores, the worse the anxiety. For depression levels, the research tools used were Profile of Mood States, Depression, Anxiety, Stress-Scale 21, and Beck Depression Inventory-II. The higher the scores, the worse the depression [41].

The other secondary outcomes, including restorative sleep, were measured by a Restorative Sleep Questionnaire (RSQ). It was evaluated by one study that reported two results [25]. The RSQ is a validated 11-item questionnaire with good psychometrics that assess the refreshing quality of sleep by asking participants to rate it on a 5-point scale. It measures feelings of tiredness, mood, and energy. The RSQ could distinguish between healthy controls, patients with primary insomnia, and insomnia patients with isolated non-restorative sleep complaints. Higher scores indicate better restorative sleep [42]. 

For assessing the quality of life, the research tool used in this review was the 36-Item Short Form Survey (SF-36) [43]. The effect of Crocus sativus on SF-36 results was evaluated by a single study [27]. The SF-36 is an instrument that assesses health-related quality of life, consisting of two main domains: physical and mental. The physical domain is represented by the Physical Component Summary (PCS), and the mental domain is represented by the Mental Component Summary (MCS). The SF-36 measures eight scales: physical functioning, physical limitation, bodily pain, general health, vitality, social functioning, emotional limitations, and mental health. All these scales contribute to the scoring of two domains, physical and mental. High scores define a more favorable health state [43].

The assessment of the risk of bias is shown in Figure 2 and Figure 3. Figure 2 shows the proportion of studies assessed as low, high, or unclear risk of bias for each risk of bias indicator. Figure 3 shows the risk of bias indicators for individual studies.

For allocation concealment, the selection bias in five studies was unclear risk [25,26,29,30,31], and all used computer-generated randomization. Three studies had unclear risks [24,27,28]. Allocation concealment methods were described in two studies [25,26] as low risk. However, six studies did not mention allocation concealment [24,27,28,29,30,31]. All eight studies included in this review used a placebo control. Four studies described the two components, blinding of participants and blinding of outcome assessment [25,27,29,31], as low risk. In four studies, the blinding of personnel was unclear [24,26,28,30]. The blinding of outcome assessments was not described in four studies [24,27,29,31] and was categorized as unclear risk. We judged blinding of personnel and outcome assessment as an unclear risk of bias. For incomplete data, six studies were assessed as low risk for attrition bias [24,25,27,29,30,31]. The incomplete data outcome was not described in the two studies [26,28]. Thus, we judged them as presenting an unclear risk of bias. All eight studies reported the outcomes specified in their methods section [24,25,26,27,28,29,30,31] for selective reporting and were assessed as low risk for selective reporting. We detected no other potential sources of bias.

### 3.2. Insomnia Severity

There was a small reduction in insomnia severity in the Crocus sativus groups compared with the placebo groups (SMD: 0.53; 95%CI: −0.05 to 1.11; I^2^ statistic = 59%; *p* = 0.08; 2 studies, 118 participants, moderate-quality evidence) [25,29] (Figure 4, Table 2).

### 3.3. Sleep Quality

Crocus sativus increased sleep quality compared with placebos (SMD: 0.89; 95%CI: 0.10 to 1.68; I^2^ statistic = 90%; *p* = 0.03; 6 studies, 308 participants, very low-quality evidence) [25,26,27,28,29,30]. (Figure 5, Table 2). 

Subgroup analysis according to comorbidity was performed. Those with diabetes mellitus showed no difference in sleep quality (SMD: 1.95; 95%CI: 0.64 to 3.25; I^2^ statistic = 87%; *p* = 0.003; 2 studies; 110 participants, very low-quality evidence) [28,30], while those without diabetes mellitus showed a reduction in sleep quality in the Crocus sativus groups compared with the placebo groups (SMD: 0.43; 95%CI: −0.01 to 0.87; I^2^ statistic = 33%; *p* = 0.06; 3 studies; 130 participants, moderate-quality evidence) [26,27,29].

Subgroup analysis by Crocus sativus preparation cannot be performed due to a limited number of trials.

### 3.4. Sleep Latency

There was no difference in sleep latency between the Crocus sativus groups and the placebo groups (SMD: 0.10; 95%CI: −0.19 to 0.38; I^2^ statistic = 0%; *p* = 0.51; 4 studies; 190 participants, low-quality evidence) [26,27,29,30] (Figure 6, Table 2). 

### 3.5. Sleep Duration

Crocus sativus increased sleep duration compared with the placebos in 5 trials (SMD: 0.57; 95%CI: 0.21 to 0.93; I^2^ statistic = 40%; *p* = 0.002; 5 studies; 220 participants, moderate-quality evidence) [26,27,29,30,31] (Figure 7, Table 2). 

### 3.6. Sleep Efficiency

There was no difference in sleep efficiency between the Crocus sativus groups and the placebo groups in 5 trials (SMD: 0.08; 95%CI: −0.18 to 0.34; I^2^ statistic = 0%; *p* = 0.54; 5 studies; 233 participants, low-quality evidence) [24,27,29,30,31] (Figure 8, Table 2). 

### 3.7. Sleep Disturbances

There was no difference in sleep disturbances between the Crocus sativus groups and the placebo groups in 3 trials (MD: 0.01, 95%CI: −0.20 to 0.21; I^2^ statistic = 0%; *p* = 0.96; 3 studies; 140 participants, moderate-quality evidence) [26,27,30] (Figure 9).

### 3.8. Use of Sleep Medications

There was no difference in the usage of sleep medications between the Crocus sativus groups and the placebo groups in 4 studies (MD: 0.14; 95%CI: −0.03 to 0.32; I^2^ statistic = 0%; *p* = 0.11; 4 studies; 190 participants, moderate-quality evidence) [26,27,29,30] (Figure 10, Table 2). 

### 3.9. Daytime Dysfunction

There was no difference in the daytime dysfunction between the Crocus sativus groups and the placebo groups in 4 studies (MD: 0.23; 95%CI: −0.14 to 0.60; I^2^ statistic = 83%; *p* = 0.22; 4 studies; 190 participants, moderate-quality evidence) [26,27,29,30].

### 3.10. Anxiety Level

There was no difference in anxiety level between the Crocus sativus group and the placebo group in 2 trials (SMD: −0.19; 95%CI: −0.76 to 0.38; I^2^ statistic = 0%; *p* = 0.52; 2 studies; 51 participants, low-quality evidence) [26,28] (Figure 11, Table 2). 

### 3.11. Depression Level

There was no difference in depression level between the Crocus sativus group and the placebo group in 3 studies (SMD: 0.41; 95%CI: −0.42 to 1.24; I^2^ statistic = 82%; *p* = 0.34; 3 studies; 149 participants, very low-quality evidence) [25,26,30] (Figure 12).

### 3.12. Restorative Sleep

Crocus sativus increased restorative sleep compared with the placebo in 1 study (MD: 4.18; 95%CI: 2.85 to 5.51; 1 study; 68 participants, very low-quality evidence) [25]. 

### 3.13. Quality of Life

There was no difference in quality of life in terms of the two domains, physical (MD: 0.81; 95%CI: −8.96 to 10.58; I^2^ statistic = 0%; *p* = 0.85; 1 study; 59 participants, very low-quality evidence) and mental (MD: 3.10; 95%CI: −19.07 to 25.27; I^2^ statistic = 0%; *p* = 0.85; 1 study; 59 participants, very low-quality evidence) [27].

## 4. Discussion

This review aimed to evaluate the effectiveness of Crocus sativus for treating insomnia in adults. We included 8 studies, 2 to 8 weeks long, with 494 participants aged from 18 to 70 years old, and the dosage of Crocus sativus ranged from 0.6 mg to 300 mg. This review showed reduced insomnia severity and increased sleep quality and duration with Crocus sativus compared with the placebos. However, there was no difference between Crocus sativus and placebo for sleep latency, sleep efficiency, sleep disturbances, use of sleep medications, daytime dysfunction, anxiety level, and depression outcomes. Crocus sativus may improve restorative sleep compared with the placebos, but the evidence is very uncertain due to a limited number of trials. No difference was reported in quality of life between the Crocus sativus and placebo groups. The consumption of Crocus sativus has anti-inflammatory effects. It may also be associated with sleep enhancement, as insomnia is associated with increased inflammatory markers. The hypnotic effects in Crocus sativus increase NREM sleep duration and decrease its latency in animal models [11]. 

The subanalysis for comorbidity could not adequately explain the high sleep quality heterogeneity. The studies among those with diabetes showed consistent findings. In addition to insomnia, most diabetic patients reported factors such as pain in the shin, muscle cramp, increased and decreased blood sugar level, and repetitive urination during the nighttime that affects sleep quality [44]. 

We comprehensively searched electronic databases to ensure we identified all relevant studies. We also screened the reference lists of all the identified studies to look for any further relevant studies. Two authors reviewed the studies for inclusion and exclusion, extracted data independently, and examined the quality of the studies. The studies included in this review were conducted in various countries, enabling applicability to various settings. However, the applicability of the findings were limited by the duration and dosage of the Crocus sativus administered in the trials.

To reduce publication bias, we searched numerous databases without regard for the publication date and reviewed the reference lists of all relevant articles for additional references. However, we could not confirm whether we had discovered all the pertinent trials in this area. Inadequate trials for each outcome also prevented us from constructing a funnel plot to assess bias and heterogeneity. 

To date, limited systematic reviews and meta-analyses have explicitly examined the effect of Crocus sativus on sleep outcomes in human trials. Most research on the efficacy of Crocus sativus has been conducted in animal models. One review that used animal models as subjects concluded that Crocus sativus reduced anxiety and increased sleep time in mice [45], and that it has anxiolytic and hypnotic effects [11]. Studies on human subjects show that Crocus sativus reduced sleep disorders in postmenopausal women [46]. Menopausal syndromes are always associated with symptoms of hot flushes, insomnia, headache, fatigue, depression, and irritability [47]. These symptoms are always associated with hormonal changes during a menopausal state [48]. Thus, the consumption of aphrodite, composed of several plants, such as ginger, saffron, cinnamon, and Tribulus Terrestris improves menopausal symptoms [46].

### 4.1. Implications for Clinical Practice

Presently, moderate- to very low-quality evidence shows that taking Crocus sativus may benefit people with insomnia. There is not much evidence regarding the adverse effects of Crocus sativus on insomnia; thus, the severity of dependency and withdrawal are not known. The findings of the included studies pertain only to short-term use of Crocus sativus (in low doses), with the longest intervention time being eight weeks. This non-pharmacological intervention may reduce the chance of adults with insomnia taking sedative–hypnotic medication, thus reducing dependency and withdrawal symptoms. 

### 4.2. Implications for Future Research

There are various methods to quantify sleep parameters, including subjective and objective measures, yet there is no consensus on the best method to use. A specific method or tool for gathering data for future studies is suggested. Though the gold standard and objective measure for sleep assessment is polysomnography, it requires participants to sleep in the laboratory or to be connected to portable polysomnography devices at home. This is impractical and burdens the participants [49]. None of the trials in this review used polysomnography to assess sleep. Alternative tools for objective sleep measures include actigraphy, observation, bed sensors, eyelid movement- and non-invasive arm sensors, a sleep switch, and a remote device [49]. An actigraph device is a non-invasive tool capable of diagnosing circadian rhythm sleep–wake disorders. It is a low-cost and convenient wearable device [50,51]. It does not directly measure sleep, but movement, which is then used to estimate sleep/wake cycles. Only one trial [27] in this review used actigraphy. More studies on humans are necessary. We have assessed adverse events, but very few of the included studies reported sufficient details.

## 5. Conclusions

This systematic review and meta-analysis found limited and very low- to moderate-quality evidence for Crocus sativus benefitting people with insomnia.

## Figures and Tables

**Figure 1 ijerph-19-11658-f001:**
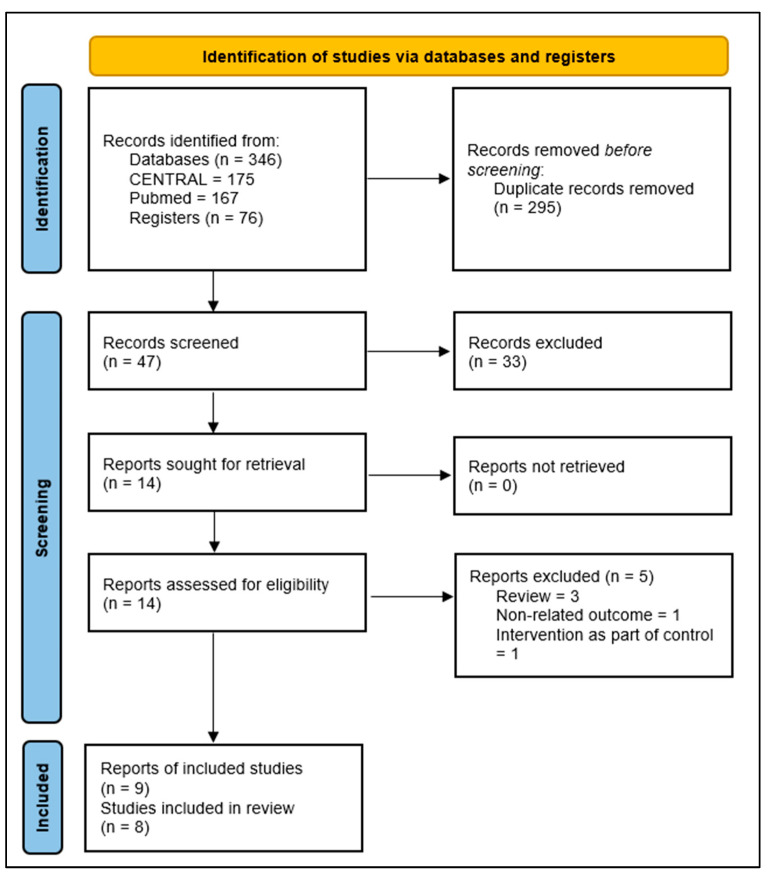
PRISMA flow chart.

**Figure 2 ijerph-19-11658-f002:**
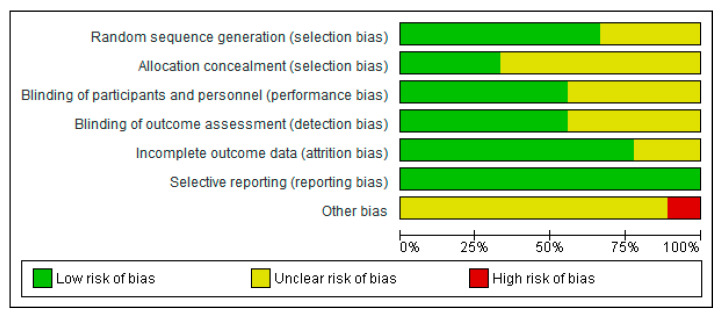
Judgement about each risk of bias item presented as percentages across all included studies.

**Figure 3 ijerph-19-11658-f003:**
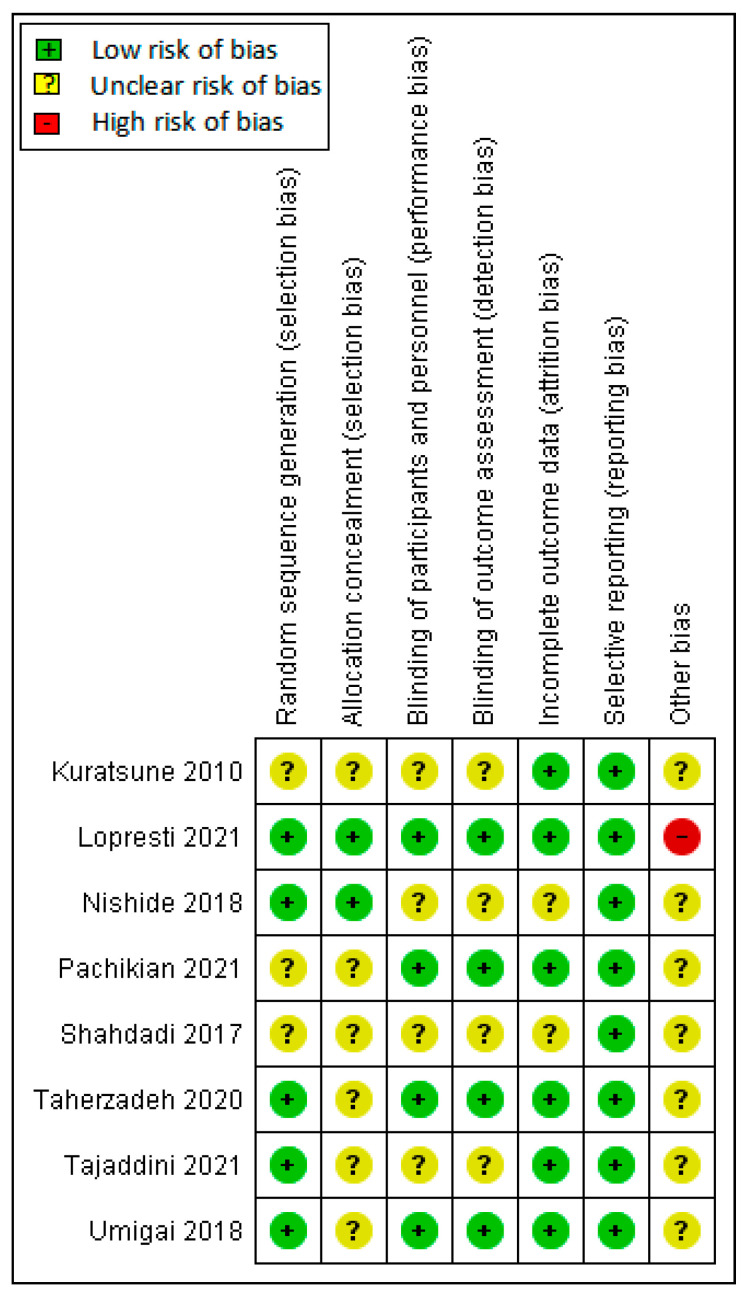
Judgements about each risk of bias item for each included study [24,25,26,27,28,29,30,31].

**Figure 4 ijerph-19-11658-f004:**
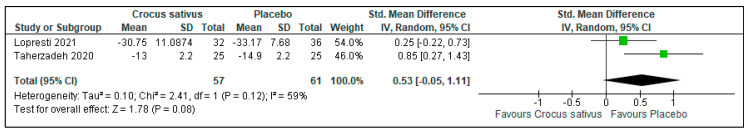
Forest plot for the outcome insomnia severity [25,29]. Note: Line indicates confidence interval and the green square indicates effect estimate. Black diamond indicates cumulative effect estimate and its confidence interval.

**Figure 5 ijerph-19-11658-f005:**
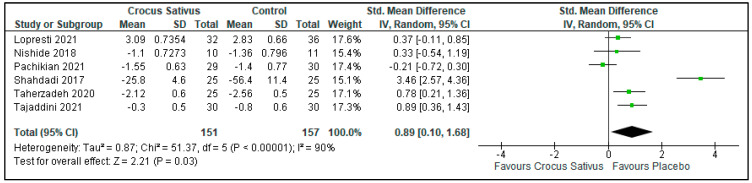
Forest plot for the outcome sleep quality [25,26,27,28,29,30]. Note: Line indicates confidence interval and the green square indicates effect estimate. Black diamond indicates cumulative effect estimate and its confidence interval.

**Figure 6 ijerph-19-11658-f006:**
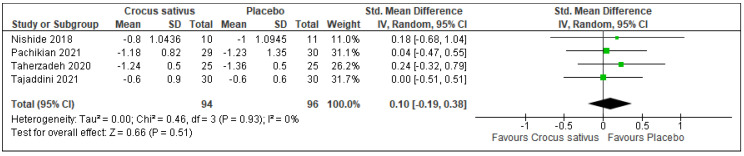
Forest plot for the outcome sleep latency [26,27,29,30]. Note: Line indicates confidence interval and the green square indicates effect estimate. Black diamond indicates cumulative effect estimate and its confidence interval.

**Figure 7 ijerph-19-11658-f007:**
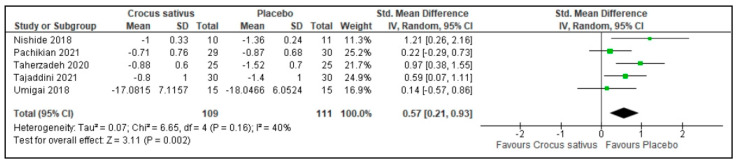
Forest plot for the outcome sleep duration [26,27,29,30,31]. Note: Line indicates confidence interval and the green square indicates effect estimate. Black diamond indicates cumulative effect estimate and its confidence interval.

**Figure 8 ijerph-19-11658-f008:**
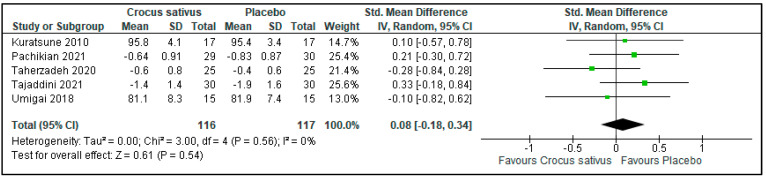
Forest plot for the outcome sleep efficiency [24,27,29,30,31]. Note: Line indicates confidence interval and the green square indicates effect estimate. Black diamond indicates cumulative effect estimate and its confidence interval.

**Figure 9 ijerph-19-11658-f009:**
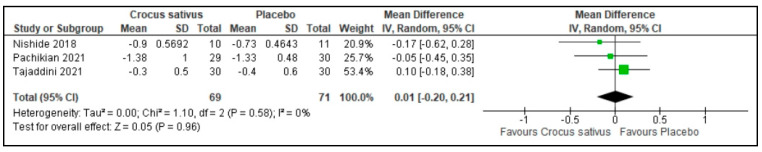
Forest plot for the outcome sleep disturbances [26,27,30]. Note: Line indicates confidence interval and the green square indicates effect estimate. Black diamond indicates cumulative effect estimate and its confidence interval.

**Figure 10 ijerph-19-11658-f010:**
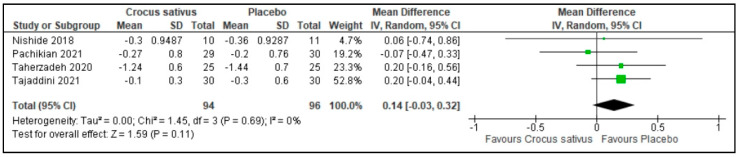
Forest plot for the outcome use of sleep medications [26,27,29,30]. Note: Line indicates confidence interval and the green square indicates effect estimate. Black diamond indicates cumulative effect estimate and its confidence interval.

**Figure 11 ijerph-19-11658-f011:**
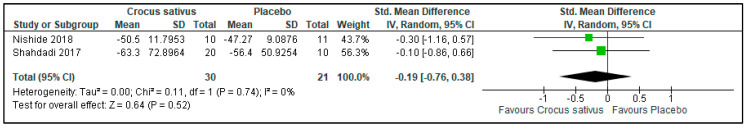
Forest plot for the outcome anxiety level [26,28]. Note: Line indicates confidence interval and the green square indicates effect estimate. Black diamond indicates cumulative effect estimate and its confidence interval.

**Figure 12 ijerph-19-11658-f012:**
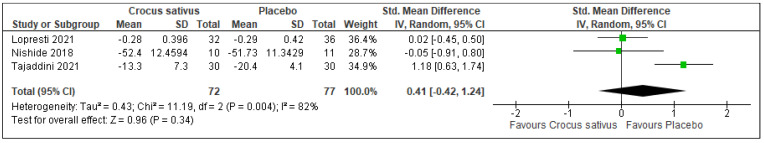
Forest plot for the outcome depression level [25,26,30]. Note: Line indicates confidence interval and the green square indicates effect estimate. Black diamond indicates cumulative effect estimate and its confidence interval.

**Table 1 ijerph-19-11658-t001:** Characteristics of included studies.

Author (Year)	Sample Size	Patient Characteristics	CS Form/Route/Daily Dose	Control	Follow-Up
Kuratsune et al. (2010) [24]	I: 17 C: 17	Healthy adults with insomnia (PSQIG ≧ 6)	I: Extract in capsule/Oral/7.5 mg	Placebo capsule (dextrin)	6 weeks
Lopresti et al. (2021) [25]	I₁: 40 I₂: 40 C: 40	Healthy adults with insomnia	I: Extract in capsule/Oral/14 mg and 28 mg	Placebo capsule (cellulose and calcium)	4 weeks
	I: 33 C: 30				
Nishide et al. (2018) [26]	I: 10 C: 11	Healthy adults	I: Extract in capsule/Oral/0.6 mg	Placebo capsule (cellulose, starch, calcium)	4 weeks
Pachikian et al. (2021) [27]	I: 34 C: 32	Healthy adults with insomnia (ISI 7-21)	I: Extract in capsule/Oral/15.5 mg	Placebo capsule (dextrin)	6 weeks
Shahdadi et al. (2017) [28]	I: 25 C: 25	Diabetes Mellitus with insomnia	I: Extract in capsule/Oral/300 mg	Placebo capsule	1 week
Taherzadeh et al. (2020) [29]	I: 25 C: 25	Healthy adults with insomnia	I: Extract in liquid/Intranasal/0.02 mg/mL	Placebo (sesame oil)	8 weeks
Tajaddini et al. (2021) [30]	I: 35 C: 35	Diabetes Mellitus with insomnia	I: Extract in capsule/Oral/100 mg	Placebo capsule (dextrin)	8 weeks
Umigai et al. (2018) [31]	I: 15 C: 15	Healthy adults with insomnia	I: Extract in capsule/Oral/30 mg	Placebo capsule (dextrin)	14 days

Note: C = Control, CS = Crocus sativus, I = Intervention, ISI = Insomnia Severity Index, PSQIG = Pittsburgh Sleep Quality Global Index score.

**Table 2 ijerph-19-11658-t002:** Summary of findings by GRADE assessment for the comparison between Crocus sativus and placebos in insomnia.

Outcomes	No. of Participants (Studies)	No. of Participants		Anticipated Absolute Effects			Certainty of Evidence (GRADE)	Comments
		Placebo	CS	MD	95%CI	*p*-Value		
Insomnia severity	118 (2 RCT)	61	57	0.53 higher	−0.05 to 1.11	0.08	⨁⨁⨁◯ Moderate	Risk of bias: not serious Inconsistency: not serious Indirectness: not serious Imprecision: serious
Sleep quality	308 (6 RCT)	157	151	0.89 higher	0.10 to 1.68	0.03	⨁◯◯◯ Very low	Risk of bias: not serious Inconsistency: serious Indirectness: serious Imprecision: serious
Sleep latency	190 (4 RCT)	96	94	0.10 higher	−0.19 to 0.38	0.51	⨁⨁◯◯ Low	Risk of bias: not serious Inconsistency: not serious Indirectness: serious Imprecision: serious
Sleep efficiency	233 (5 RCT)	117	116	0.08 higher	−0.18 to 0.34	0.54	⨁⨁◯◯ Low	Risk of bias: not serious Inconsistency: not serious Indirectness: serious Imprecision: serious
Sleep duration	220 (5 RCT)	111	109	0.57 higher	0.21 to 0.93	0.002	⨁⨁⨁◯ Moderate	Risk of bias: not serious Inconsistency: not serious Indirectness: not serious Imprecision: serious
Use of sleep medications	190 (4 RCT)	96	94	0.14 higher	−0.03 to 0.32	0.11	⨁⨁⨁◯ Moderate	Risk of bias: not serious Inconsistency: not serious Indirectness: not serious Imprecision: serious
Anxiety Level	51 (2 RCT)	21	30	0.19 lower	−0.76 to 0.38	0.52	⨁⨁◯◯ Low	Risk of bias: not serious Inconsistency: not serious Indirectness: serious Imprecision: serious

Note: ⨁ indicates the certainty level.

## Data Availability

Data are contained within the article.

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
