# Peer review of "Crocus Sativus for Insomnia: A Systematic Review and Meta-Analysis"

_ijerph, 2022, doi:10.3390/ijerph191811658_

Round 1

Reviewer 1 Report

Insomnia is a world-wide public health problem which needs globe attention to it. Herbal medicine has been used for treatment for insomnia, while the effect remains debatable. This review entitled” Crocus sativus for insomnia: a systematic review and meta-analysis” summarized publications that report the effect of Crocus sativus on insomnia in adults. The authors then applied meta-analysis to assess the function of Crocus sativus for sleep quality. This is an interesting review which is important to both biomedical and clinic research community. The review is clear structured and well written. I only have the following suggestions to make it better prior to publication:

1.  Line 13: should be “31 May 2022”

2.  More detailed figure legend is required for Figure 3 to explain the figure. For example, what do the colors indicate? What does the question mark mean? What does the minus sign mean?

Author Response

As in the attachment. Thank you

Reviewer 2 Report

Abstract, line 11: “This study aimed”, not “This aimed”.

Abstract, line 12: “A systematic search of PubMed/Medline and the Cochrane Library was undertaken for literature through August 2021”, instead of the current.

Abstract, line 20: Recommend replacing “Crocus sativus may be beneficial for insomnia”, with “Although there remains only limited and very low to moderate quality evidence that taking Crocus sativus may benefit people with insomnia, this non-pharmacological intervention MAY reduce the chance…”

Introduction, page 1, line 26: “Insomnia may be experienced in”, not “Insomnia always happens at”.

Introduction, page 2, line 62: Eliminate “a” in “including a diuretic”.

Introduction, page 2, line 68: Delete the sentence beginning “The active ingredient comprised”, as this is redundant.

Materials and Methods, page 2, line 94: There is an inappropriate carriage return after “Cochrane Central” and “Register of Controlled Trials”.

Materials and Methods, page 2, line 96: Since the databases were current only through August 2021, the search should be specified as being through this date. 

Materials and Methods, page 2, line 104: “We adapted the search strategy for other databases”. These are not listed. If this refers only to the WHO ICTRP, this sentence should be listed at the end, referencing only this database. 

Figure 2: The caption to this figure should make clear that the proportions shown refer to proportions of the studied papers. Perhaps “Risk of bias in the assessed studies”

Implications for clinical practice, page 14, line 441, replace “probably” with “may”.

Conclusions: This section is unsatisfactory and is at odds with the abstract. Recommend “This systematic review finds only limited and very low to moderate quality evidence that taking Crocus sativus may benefit people with insomnia.”

Author Response

As in the attachment. Thank you
